# A Scalable Approach for Privacy-Preserving Collaborative Machine Learning

**J. So**
ECE Department
University of Southern California (USC)
jinhyuns@usc.edu

**B. Guler**
ECE Department
University of California, Riverside
bguler@ece.ucr.edu

**A.S. Avestimehr**
ECE Department
University of Southern California (USC)
avestimehr@ee.usc.edu

## Abstract

We consider a collaborative learning scenario in which multiple data-owners wish to jointly train a logistic regression model, while keeping their individual datasets private from the other parties. We propose COPML, a fully-decentralized training framework that achieves scalability and privacy-protection simultaneously. The key idea of COPML is to securely encode the individual datasets to distribute the computation load effectively across many parties and to perform the training computations as well as the model updates in a distributed manner on the securely encoded data. We provide the privacy analysis of COPML and prove its convergence. Furthermore, we experimentally demonstrate that COPML can achieve significant speedup in training over the benchmark protocols. Our protocol provides strong statistical privacy guarantees against colluding parties (adversaries) with unbounded computational power, while achieving up to $16\times$ speedup in the training time against the benchmark protocols.

## 1 Introduction

Machine learning applications can achieve significant performance gains by training on large volumes of data. In many applications, the training data is distributed across multiple data-owners, such as patient records at multiple medical institutions, and furthermore contains sensitive information, e.g., genetic information, financial transactions, and geolocation information. Such settings give rise to the following key problem that is the focus of this paper: *How can multiple data-owners jointly train a machine learning model while keeping their individual datasets private from the other parties?*

More specifically, we consider a distributed learning scenario in which $N$ data-owners (clients) wish to train a logistic regression model jointly without revealing information about their individual datasets to the other parties, even if up to $T$ out of $N$ clients collude. Our focus is on the semi-honest adversary setup, where the corrupted parties follow the protocol but may leak information in an attempt to learn the training dataset. To address this challenge, we propose a novel framework, COPML[1], that enables fast and privacy-preserving training by leveraging information and coding theory principles. COPML has three salient features:

- speeds up the training time significantly, by distributing the computation load effectively across a large number of parties,

- advances the state-of-the-art privacy-preserving training setups by scaling to a large number of parties, as it can distribute the computation load effectively as more parties are added in the system,
- utilizes coding theory principles to secret share the dataset and model parameters which can significantly reduce the communication overhead and the complexity of distributed training.

At a high level, COPML can be described as follows. Initially, the clients secret share their individual datasets with the other parties, after which they carry out a secure multi-party computing (MPC) protocol to *encode* the dataset. This encoding operation transforms the dataset into a *coded* form that enables faster training and simultaneously guarantees privacy (in an information-theoretic sense). Training is performed over the encoded data via gradient descent. The parties perform the computations over the encoded data *as if they were computing over the uncoded dataset*. That is, the structure of the computations are the same for computing over the uncoded dataset versus computing over the encoded dataset. At the end of training, each client should only learn the final model, and no information should be leaked (in an information-theoretic sense) about the individual datasets or the intermediate model parameters, beyond the final model.

We characterize the theoretical performance guarantees of COPML, in terms of convergence, scalability, and privacy protection. Our analysis identifies a trade-off between privacy and parallelization, such that, each additional client can be utilized either for more privacy, by protecting against a larger number of collusions $T$, or more parallelization, by reducing the computation load at each client. Furthermore, we empirically demonstrate the performance of COPML by comparing it with cryptographic benchmarks based on secure multi-party computing (MPC) [39, 4, 3, 12], that can also be applied to enable privacy-preserving machine learning tasks (e.g. see [30, 14, 28, 25, 10, 8, 37, 27]). Given our focus on information-theoretic privacy, the most relevant MPC-based schemes for empirical comparison are the protocols from [4] and [3, 12] based on Shamir's secret sharing [33]. While several more recent works have considered MPC-based learning setups with information-theoretic privacy [37, 27], their constructions are limited to three or four parties.

We run extensive experiments over the Amazon EC2 cloud platform to empirically demonstrate the performance of COPML. We train a logistic regression model for image classification over the CIFAR-10 [23] and GISETTE [18] datasets. The training computations are distributed to up to $N = 50$ parties. We demonstrate that COPML can provide significant speedup in the training time against the state-of-the-art MPC baseline (up to $16.4\times$), while providing comparable accuracy to conventional logistic regression. This is primarily due to the parallelization gain provided by our system, which can distribute the workload effectively across many parties.

**Other related works.** Other than MPC-based setups, one can consider two notable approaches. The first one is Homomorphic Encryption (HE) [15], which enables computations on encrypted data, and has been applied to privacy-preserving machine learning [16, 20, 17, 41, 24, 22, 38, 19]. The privacy protection of HE depends on the size of the encrypted data, and computing in the encrypted domain is computationally intensive. The second approach is differential privacy (DP), which is a noisy release mechanism to protect the privacy of personally identifiable information. The main application of DP in machine learning is when the model is to be released publicly after training, so that individual data points cannot be backtracked from the released model [7, 34, 1, 31, 26, 32, 21]. On the other hand, our focus is on ensuring privacy during training, while preserving the accuracy of the model.

## 2 Problem Setting

We consider a collaborative learning scenario in which the training dataset is distributed across $N$ clients. Client $j \in [N]$ holds an individual dataset denoted by a matrix $\mathbf{X}_j \in \mathbb{R}^{m_j \times d}$ consisting of $m_j$ data points with $d$ features, and the corresponding labels are given by a vector $\mathbf{y}_j \in \{0, 1\}^{m_j}$. The overall dataset is denoted by $\mathbf{X} = [\mathbf{X}_1^\top, \dots, \mathbf{X}_N^\top]^\top$ consisting of $m \triangleq \sum_{j \in [N]} m_j$ data points with $d$ features, and corresponding labels $\mathbf{y} = [\mathbf{y}_1^\top, \dots, \mathbf{y}_N^\top]^\top$, which consists of $N$ individual datasets each one belonging to a different client. The clients wish to jointly train a logistic regression model $\mathbf{w}$ over the training set $\mathbf{X}$ with labels $\mathbf{y}$, by minimizing a cross entropy loss function,

$$C(\mathbf{w}) = \frac{1}{m} \sum_{i=1}^{m} \left( -y_i \log \hat{y}_i - (1 - y_i) \log(1 - \hat{y}_i) \right) \tag{1}$$

where $\hat{y}_i = g(\mathbf{x}_i \cdot \mathbf{w}) \in (0, 1)$ is the probability of label $i$ being equal to 1, $\mathbf{x}_i$ is the $i^{th}$ row of matrix $\mathbf{X}$, and $g(\cdot)$ denotes the sigmoid function $g(z) = 1/(1 + e^{-z})$. The training is performed through gradient descent, by updating the model parameters in the opposite direction of the gradient,

$$\mathbf{w}^{(t+1)} = \mathbf{w}^{(t)} - \frac{\eta}{m}\mathbf{X}^\top(g(\mathbf{X} \times \mathbf{w}^{(t)}) - \mathbf{y}) \tag{2}$$

where $\nabla C(\mathbf{w}) = \frac{1}{m}\mathbf{X}^\top(g(\mathbf{X} \times \mathbf{w}) - \mathbf{y})$ is the gradient for (1), $\mathbf{w}^{(t)}$ holds the estimated parameters from iteration $t$, $\eta$ is the learning rate, and function $g(\cdot)$ acts element-wise over the vector $\mathbf{X} \times \mathbf{w}^{(t)}$.

During training, the clients wish to protect the privacy of their individual datasets from other clients, even if up to $T$ of them collude, where $T$ is the *privacy parameter* of the system. There is no trusted party who can collect the datasets in the clear and perform the training. Hence, the training protocol should preserve the privacy of the individual datasets against any collusions between up to $T$ adversarial clients. More specifically, this condition states that the adversarial clients should not learn any information about the datasets of the benign clients beyond what can already be inferred from the adversaries' own datasets.

To do so, client $j \in [N]$ initially secret shares its individual dataset $\mathbf{X}_j$ and $\mathbf{y}_j$ with the other parties. Next, clients carry out a secure MPC protocol to encode the dataset by using the received secret shares. In this phase, the dataset $\mathbf{X}$ is first partitioned into $K$ submatrices $\mathbf{X} = [\mathbf{X}_1^\top, \cdots, \mathbf{X}_K^\top]^\top$ for some $K \in \mathbb{N}$. Parameter $K$ characterizes the computation load at each client. Specifically, our system ensures that the computation load (in terms of gradient computations) at each client is equal to processing only $(1/K)^{th}$ of the entire dataset $\mathbf{X}$. The clients then encode the dataset by combining the $K$ submatrices together with some randomness to preserve privacy. At the end of this phase, client $i \in [N]$ learns an encoded dataset $\widetilde{\mathbf{X}}_i$, whose size is equal to $(1/K)^{th}$ of the dataset $\mathbf{X}$. This process is only performed once for the dataset $\mathbf{X}$.

At each iteration of training, clients also encode the current estimation of the model parameters $\mathbf{w}^{(t)}$ using a secure MPC protocol, after which client $i \in [N]$ obtains the encoded model $\widetilde{\mathbf{w}}_i^{(t)}$. Client $i \in [N]$ then computes a local gradient $\widetilde{\mathbf{X}}_i^\top g(\widetilde{\mathbf{X}}_i \times \widetilde{\mathbf{w}}_i^{(t)})$ over the encoded dataset $\widetilde{\mathbf{X}}_i$ and encoded model $\widetilde{\mathbf{w}}_i^{(t)}$. After this step, clients carry out another secure MPC protocol to decode the gradient $\mathbf{X}^\top g(\mathbf{X} \times \mathbf{w}^{(t)})$ and update the model according to (2). As the decoding and model updates are performed using a secure MPC protocol, clients do not learn any information about the actual gradients or the updated model. In particular, client $i \in [N]$ only learns a secret share of the updated model, denoted by $[\mathbf{w}^{(t+1)}]_i$. Using the secret shares $[\mathbf{w}^{(t+1)}]_i$, clients $i \in [N]$ encode the model $\mathbf{w}^{(t+1)}$ for the next iteration, after which client $i$ learns an encoded model $\widetilde{\mathbf{w}}_i^{(t+1)}$. Figure 1 demonstrates our system architecture.

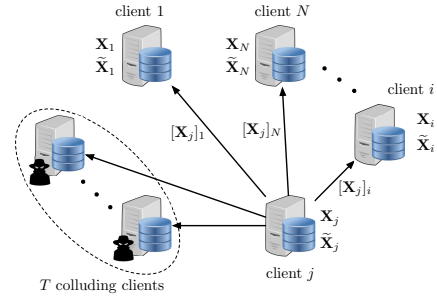

Figure 1: The multi-client distributed training setup with $N$ clients. Client $j \in [N]$ holds a dataset $\mathbf{X}_j$ with labels $\mathbf{y}_j$. At the beginning of training, client $j$ secret shares $\mathbf{X}_j$ and $\mathbf{y}_j$ to guarantee their information-theoretic privacy against any collusions between up to $T$ clients. The secret share of $\mathbf{X}_j$ and $\mathbf{y}_j$ assigned from client $j$ to client $i$ is represented by $[\mathbf{X}_j]_i$ and $[\mathbf{y}_j]_i$, respectively.

## 3  The COPML Framework

COPML consists of four main phases: quantization; encoding and secret sharing; polynomial approximation; decoding and model update, as demonstrated in Figure 2. In the first phase, quantization, each client converts its own dataset from the real domain to finite field. In the second phase, clients create a secret share of their quantized datasets and carry out a secure MPC protocol to encode the datasets. At each iteration, clients also encode and create a secret share of the model parameters. In the third phase, clients perform local gradient computations over the encoded datasets and encoded model parameters by approximating the sigmoid function with a polynomial. Then, in the last phase, clients decode the local computations and update the model parameters using a secure MPC protocol. This process is repeated until the convergence of the model parameters.

**Phase 1: Quantization.** Computations involving secure MPC protocols are bound to finite field operations, which requires the representation of real-valued data points in a finite field $\mathbb{F}$. To do so, each client initially quantizes its dataset from the real domain to the domain of integers, and then

embeds it in a field $\mathbb{F}_p$ of integers modulo a prime $p$. Parameter $p$ is selected to be sufficiently large to avoid wrap-around in computations. For example, in a 64-bit implementation with the CIFAR-10 dataset, we select $p = 2^{26} - 5$. The details of the quantization phase are provided in Appendix A.1.

**Phase 2: Encoding and secret sharing.** In this phase, client $j \in [N]$ creates a secret share of its quantized dataset $\mathbf{X}_j$ designated for each client $i \in [N]$ (including client $j$ itself). The secret shares are constructed via Shamir's secret sharing with threshold $T$ [33], to protect the privacy of the individual datasets against any collusions between up to $T$ clients. To do so, client $j$ creates a random polynomial, $h_j(z) = \mathbf{X}_j + z\mathbf{R}_{j1} + \ldots + z^T\mathbf{R}_{jT}$ where $\mathbf{R}_{ji}$ for $i \in [T]$ are i.i.d. uniformly distributed random matrices, and selects $N$ distinct evaluation points $\lambda_1, \ldots, \lambda_N$ from $\mathbb{F}_p$. Then, client $j$ sends

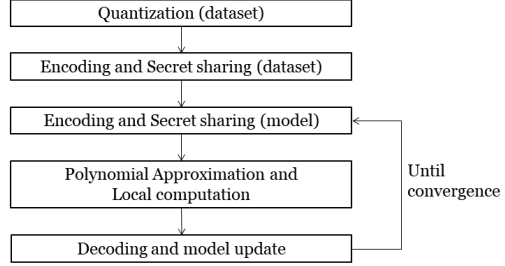

Figure 2: Flowchart of COPML.

client $i \in [N]$ a secret share $[\mathbf{X}_j]_i \triangleq h_j(\lambda_i)$ of its dataset $\mathbf{X}_j$. Client $j$ also sends a secret share of its labels $\mathbf{y}_j$ to client $i \in [N]$, denoted by $[\mathbf{y}_j]_i$. Finally, the model is initialized randomly within a secure MPC protocol between the clients, and at the end client $i \in [N]$ obtains a secret share $[\mathbf{w}^{(0)}]_i$ of the initial model $\mathbf{w}^{(0)}$.

After obtaining the secret shares $[\mathbf{X}_j]_i$ for $j \in [N]$, clients $i \in [N]$ encode the dataset using a secure MPC protocol and transform it into a *coded* form, which speeds up the training by distributing the computation load of gradient evaluations across the clients. Our encoding strategy utilizes Lagrange coding from [40][2], which has been applied to other problems such as privacy-preserving offloading of a training task [36] and secure federated learning [35]. However, we encode (and later decode) the secret shares of the datasets and not their true values. Therefore, clients do not learn any information about the true value of the dataset $\mathbf{X}$ during the encoding-decoding process.

The individual steps of the encoding process are as follows. Initially, the dataset $\mathbf{X}$ is partitioned into $K$ submatrices $\mathbf{X} = [\mathbf{X}_1^\top, \ldots, \mathbf{X}_K^\top]^\top$ where $\mathbf{X}_k \in \mathbb{F}_p^{\frac{m}{K} \times d}$ for $k \in [K]$. To do so, client $i \in [N]$ locally concatenates $[\mathbf{X}_j]_i$ for $j \in [N]$ and partitions it into $K$ parts, $[\mathbf{X}_k]_i$ for $k \in [K]$. Since this operation is done over the secret shares, clients do not learn any information about the original dataset $\mathbf{X}$. Parameter $K$ quantifies the computation load at each client, as will be discussed in Section 4.

The clients agree on $K + T$ distinct elements $\{\beta_k\}_{k \in [K+T]}$ and $N$ distinct elements $\{\alpha_i\}_{i \in [N]}$ from $\mathbb{F}_p$ such that $\{\alpha_i\}_{i \in [N]} \cap \{\beta_k\}_{k \in [K+T]} = \varnothing$. Client $i \in [N]$ then encodes the dataset using a Lagrange interpolation polynomial $u : \mathbb{F}_p \to \mathbb{F}_p^{\frac{m}{K} \times d}$ with degree at most $K + T - 1$,

$$[u(z)]_i \triangleq \sum_{k \in [K]} [\mathbf{X}_k]_i \cdot \prod_{l \in [K+T] \setminus \{k\}} \frac{z - \beta_l}{\beta_k - \beta_l} + \sum_{k=K+1}^{K+T} [\mathbf{Z}_k]_i \cdot \prod_{l \in [K+T] \setminus \{k\}} \frac{z - \beta_l}{\beta_k - \beta_l}, \quad (3)$$

where $[u(\beta_k)]_i = [\mathbf{X}_k]_i$ for $k \in [K]$ and $i \in [N]$. The matrices $\mathbf{Z}_k$ are generated uniformly at random[3] from $\mathbb{F}_p^{\frac{m}{K} \times d}$ and $[\mathbf{Z}_k]_i$ is the secret share of $\mathbf{Z}_k$ at client $i$. $[\mathbf{Z}_k]_i$ is the secret share of $\mathbf{Z}_k$ at client $i$. Client $i \in [N]$ then computes and sends $[\widetilde{\mathbf{X}}_j]_i \triangleq [u(\alpha_j)]_i$ to client $j \in [N]$. Upon receiving $\{[\widetilde{\mathbf{X}}_j]_i\}_{i \in [N]}$, client $j \in [N]$ can recover the encoded matrix $\widetilde{\mathbf{X}}_j$.[4] The role of $\mathbf{Z}_k$'s are to mask the dataset so that the encoded matrices $\widetilde{\mathbf{X}}_j$ reveal no information about the dataset $\mathbf{X}$, even if up to $T$ clients collude, as detailed in Section 4.

Using the secret shares $[\mathbf{X}_j]_i$ and $[\mathbf{y}_j]_i$, clients $i \in [N]$ also compute $\mathbf{X}^T\mathbf{y} = \sum_{j \in [N]} \mathbf{X}_j^T\mathbf{y}_j$ using a secure multiplication protocol (see Appendix A.3 for details). At the end of this step, clients learn a secret share of $\mathbf{X}^T\mathbf{y}$, which we denote by $[\mathbf{X}^T\mathbf{y}]_i$ for client $i \in N$.

At iteration $t$, client $i$ initially holds a secret share of the current model, $[\mathbf{w}^{(t)}]_i$, and then encodes the model via a Lagrange interpolation polynomial $v : \mathbb{F}_p \to \mathbb{F}_p^d$ with degree at most $K + T - 1$,

$$[v(z)]_i \triangleq \sum_{k \in [K]} [\mathbf{w}^{(t)}]_i \cdot \prod_{l \in [K+T] \backslash \{k\}} \frac{z - \beta_l}{\beta_k - \beta_l} + \sum_{k=K+1}^{K+T} [\mathbf{v}_k^{(t)}]_i \cdot \prod_{l \in [K+T] \backslash \{k\}} \frac{z - \beta_l}{\beta_k - \beta_l}, \qquad (4)$$

where $[v(\beta_k)]_i = [\mathbf{w}^{(t)}]_i$ for $k \in [K]$ and $i \in [N]$. The vectors $\mathbf{v}_k^{(t)}$ are generated uniformly at random from $\mathbb{F}_p^d$. Client $i \in [N]$ then sends $[\widetilde{\mathbf{w}}_j^{(t)}]_i \triangleq [v(\alpha_j)]_i$ to client $j \in [N]$. Upon receiving $\{[\widetilde{\mathbf{w}}_j^{(t)}]_i\}_{i \in [N]}$, client $j \in [N]$ recovers the encoded model $\widetilde{\mathbf{w}}_j^{(t)}$.

**Phase 3: Polynomial Approximation and Local Computations.** Lagrange encoding can be used to compute polynomial functions only, whereas the gradient computations in (2) are not polynomial operations due to the sigmoid function. To this end, we approximate the sigmoid with a polynomial,

$$\hat{g}(z) = \sum_{i=0}^{r} c_i z^i, \qquad (5)$$

where $r$ and $c_i$ represent the degree and coefficients of the polynomial, respectively. The coefficients are evaluated by fitting the sigmoid to the polynomial function via least squares estimation. Using this polynomial approximation, we rewrite the model update from (2) as,

$$\mathbf{w}^{(t+1)} = \mathbf{w}^{(t)} - \frac{\eta}{m} \mathbf{X}^{\top} (\hat{g}(\mathbf{X} \times \mathbf{w}^{(t)}) - \mathbf{y}). \qquad (6)$$

Client $i \in [N]$ then locally computes the gradient over the encoded dataset, by evaluating a function,

$$f(\widetilde{\mathbf{X}}_i, \widetilde{\mathbf{w}}_i^{(t)}) = \widetilde{\mathbf{X}}_i^{\top} \hat{g}(\widetilde{\mathbf{X}}_i \times \widetilde{\mathbf{w}}_i^{(t)}) \qquad (7)$$

and secret shares the result with the other clients, by sending a secret share of (7), $[f(\widetilde{\mathbf{X}}_i, \widetilde{\mathbf{w}}_i^{(t)})]_j$, to client $j \in [N]$. At the end of this step, client $j$ holds the secret shares $[f(\widetilde{\mathbf{X}}_i, \widetilde{\mathbf{w}}_i^{(t)})]_j$ corresponding to the local computations from clients $i \in [N]$. Note that (7) is a polynomial function evaluation in the finite field arithmetic and the degree of function $f$ is $\deg(f) = 2r + 1$.

**Phase 4: Decoding and Model Update.** In this phase, clients perform the decoding of the gradient using a secure MPC protocol, through polynomial interpolation over the secret shares $[f(\widetilde{\mathbf{X}}_i, \widetilde{\mathbf{w}}_i^{(t)})]_j$. The minimum number of clients needed for the decoding operation to be successful, which we call the *recovery threshold* of the protocol, is equal to $(2r+1)(K+T-1)+1$. In order to show this, we first note that, from the definition of Lagrange polynomials in (3) and (4), one can define a univariate polynomial $h(z) = f\big(u(z), v(z)\big)$ such that

$$h(\beta_i) = f\big(u(\beta_i), v(\beta_i)\big) = f\big(\mathbf{X}_i, \mathbf{w}^{(t)}\big) = \mathbf{X}_i^{\top} \hat{g}(\mathbf{X}_i \times \mathbf{w}^{(t)}) \qquad (8)$$

for $i \in [K]$. Moreover, from (7), we know that client $i$ performs the following computation,

$$h(\alpha_i) = f\big(u(\alpha_i), v(\alpha_i)\big) = f\big(\widetilde{\mathbf{X}}_i, \widetilde{\mathbf{w}}_i^{(t)}\big) = \widetilde{\mathbf{X}}_i^{\top} \hat{g}(\widetilde{\mathbf{X}}_i \times \widetilde{\mathbf{w}}_i^{(t)}). \qquad (9)$$

The decoding process is based on the intuition that, the computations from (9) can be used as evaluation points $h(\alpha_i)$ to interpolate the polynomial $h(z)$. Since the degree of the polynomial $h(z)$ is $\deg\big(h(z)\big) \leq (2r+1)(K+T-1)$, all of its coefficients can be determined as long as there are at least $(2r+1)(K+T-1)+1$ evaluation points available. After $h(z)$ is recovered, the computation results in (8) correspond to $h(\beta_i)$ for $i \in [K]$.

Our decoding operation corresponds to a finite-field polynomial interpolation problem. More specifically, upon receiving the secret shares of the local computations $[f(\widetilde{\mathbf{X}}_j, \widetilde{\mathbf{w}}_j^{(t)})]_i$ from at least $(2r+1)(K+T-1)+1$ clients, client $i$ locally computes

$$[f(\mathbf{X}_k, \mathbf{w}^{(t)})]_i \triangleq \sum_{j \in \mathcal{I}_i} [f(\widetilde{\mathbf{X}}_j, \widetilde{\mathbf{w}}_j^{(t)})]_i \cdot \prod_{l \in \mathcal{I}_i \backslash \{j\}} \frac{\beta_k - \alpha_l}{\alpha_j - \alpha_l} \qquad (10)$$

for $k \in [K]$, where $\mathcal{I}_i \subseteq [N]$ denotes the set of the $(2r+1)(K+T-1)+1$ fastest clients who send their secret share $[f(\widetilde{\mathbf{X}}_j, \widetilde{\mathbf{w}}_j^{(t)})]_i$ to client $i$.

After this step, client $i$ locally aggregates its secret shares $[f(\mathbf{X}_k, \mathbf{w}^{(t)})]_i$ to compute $\sum_{k=1}^K [f(\mathbf{X}_k, \mathbf{w}^{(t)})]_i$, which in turn is a secret share of $\mathbf{X}^T \hat{g}(\mathbf{X} \times \mathbf{w}^{(t)})$ since,

$$\sum_{k=1}^K f(\mathbf{X}_k, \mathbf{w}^{(t)}) = \sum_{k=1}^K \mathbf{X}_k^\top \hat{g}(\mathbf{X}_k \times \mathbf{w}^{(t)}) = \mathbf{X}^\top \hat{g}(\mathbf{X} \times \mathbf{w}^{(t)}). \tag{11}$$

Let $[\mathbf{X}^\top \hat{g}(\mathbf{X} \times \mathbf{w}^{(t)})]_i \triangleq \sum_{k=1}^K [f(\mathbf{X}_k, \mathbf{w}^{(t)})]_i$ denote the secret share of (11) at client $i$. Client $i$ then computes $[\mathbf{X}^\top \hat{g}(\mathbf{X} \times \mathbf{w}^{(t)})]_i - [\mathbf{X}^\top \mathbf{y}]_i$, which in turn is a secret share of the gradient $\mathbf{X}^\top (\hat{g}(\mathbf{X} \times \mathbf{w}^{(t)}) - \mathbf{y})$. Since the decoding operations are carried out using the secret shares, at the end of the decoding process, the clients only learn a secret share of the gradient and not its true value.

Next, clients update the model according to (6) using a secure MPC protocol, using the secret shared model $[\mathbf{w}^{(t)}]_i$ and the secret share of the gradient $[\mathbf{X}^\top \hat{g}(\mathbf{X} \times \mathbf{w}^{(t)})]_i - [\mathbf{X}^\top \mathbf{y}]_i$. A major challenge in performing the model update in (6) in the finite field is the multiplication with parameter $\frac{\eta}{m}$, where $\frac{\eta}{m} < 1$. In order to perform this operation in the finite field, one potential approach is to treat it as a computation on integer numbers and preserve full accuracy of the results. This in turn requires a very large field size as the range of results grows exponentially with the number of multiplications, which becomes quickly impractical as the number of iterations increase [28]. Instead, we address this problem by leveraging the secure truncation technique from [6]. This protocol takes secret shares $[a]_i$ of a variable $a$ as input as well as two public integer parameters $k_1$ and $k_2$ such that $a \in \mathbb{F}_{2^{k_2}}$ and $0 < k_1 < k_2$. The protocol then returns the secret shares $[z]_i$ for $i \in [N]$ such that $z = \lfloor \frac{a}{2^{k_1}} \rfloor + s$ where $s$ is a random bit with probability $P(s = 1) = (a \mod 2^{k_1})/(2^{k_1})$. Accordingly, the protocol rounds $a/(2^{k_1})$ to the closest integer with probability $1 - \tau$, with $\tau$ being the distance between $a/(2^{k_1})$ and that integer. The truncation operation ensures that the range of the updated model always stays within the range of the finite field.

Since the model update is carried out using a secure MPC protocol, at the end of this step, client $i \in [N]$ learns only a secret share $[\mathbf{w}^{(t+1)}]_i$ of the updated model $\mathbf{w}^{(t+1)}$, and not its actual value. In the next iteration, using $[\mathbf{w}^{(t+1)}]_i$, client $i \in [N]$ locally computes $[\widetilde{\mathbf{w}}_j^{(t+1)}]_i$ from (4) and sends it to client $j \in [N]$. Client $j$ then recovers the encoded model $\widetilde{\mathbf{w}}_j^{(t+1)}$, which is used to compute (7).

The implementation details of the MPC protocols are provided in Appendix A.3. The overall algorithm for COPML is presented in Appendix A.5.

## 4 Convergence and Privacy Guarantees

Consider the cost function in (1) with the quantized dataset, and denote $\mathbf{w}^*$ as the optimal model parameters that minimize (1). In this subsection, we prove that COPML guarantees convergence to the optimal model parameters (i.e., $\mathbf{w}^*$) while maintaining the privacy of the dataset against colluding clients. This result is stated in the following theorem.

**Theorem 1.** *For training a logistic regression model in a distributed system with $N$ clients using the quantized dataset $\mathbf{X} = [\mathbf{X}_1^\top, \ldots, \mathbf{X}_N^\top]^\top$, initial model parameters $\mathbf{w}^{(0)}$, and constant step size $\eta \leq 1/L$ (where $L = \frac{1}{4}\|\mathbf{X}\|_2^2$), COPML guarantees convergence,*

$$\mathbb{E}\big[C\big(\frac{1}{J}\sum_{t=0}^J \mathbf{w}^{(t)}\big)\big] - C(\mathbf{w}^*) \leq \frac{\|\mathbf{w}^{(0)} - \mathbf{w}^*\|^2}{2\eta J} + \eta \sigma^2 \tag{12}$$

*in $J$ iterations, for any $N \geq (2r+1)(K+T-1)+1$, where $r$ is the degree of the polynomial in (5) and $\sigma^2$ is the variance of the quantization error of the secure truncation protocol.*

*Proof.* The proof of Theorem 1 is presented in Appendix A.2. $\square$

As for the privacy guarantees, COPML protects the statistical privacy of the individual dataset of each client against up to $T$ colluding adversarial clients, even if the adversaries have unbounded computational power. The privacy protection of COPML follows from the fact that all building blocks of the algorithm guarantees either (strong) information-theoretic privacy or statistical privacy of the individual datasets against any collusions between up to $T$ clients. Information-theoretic privacy of Lagrange coding against $T$ colluding clients follows from [40]. Moreover, encoding, decoding, and model update operations are carried out in a secure MPC protocol that protects the

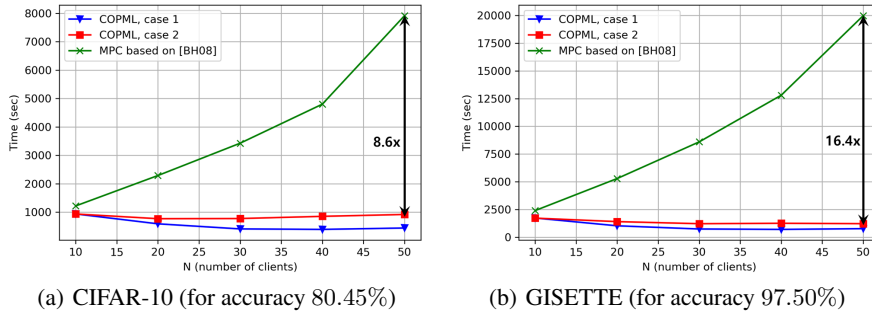

(a) CIFAR-10 (for accuracy $80.45\%$)      (b) GISETTE (for accuracy $97.50\%$)

Figure 3: Performance gain of COPML over the MPC baseline ([BH08] from [3]). The plot shows the total training time for different number of clients $N$ with 50 iterations.

information-theoretic privacy of the corresponding computations against $T$ colluding clients [4, 3, 12]. Finally, the (statistical) privacy guarantees of the truncation protocol follows from [6].

**Remark 1.** (Privacy-parallelization trade-off) Theorem 1 reveals an important trade-off between privacy and parallelization in COPML. Parameter $K$ reflects the amount of parallelization. In particular, the size of the encoded matrix at each client is equal to $(1/K)^{th}$ of the size of $\mathbf{X}$. Since each client computes the gradient over the encoded dataset, the computation load at each client is proportional to processing $(1/K)^{th}$ of the entire dataset. As $K$ increases, the computation load at each client decreases. Parameter $T$ reflects the privacy threshold of COPML. In a distributed system with $N$ clients, COPML can achieve any $K$ and $T$ as long as $N \geq (2r+1)(K+T-1)+1$. Moreover, as the number of clients $N$ increases, parallelization ($K$) and privacy ($T$) thresholds of COPML can also increase linearly, providing a scalable solution. The motivation behind the encoding process is to distribute the load of the computationally-intensive gradient evaluations across multiple clients (enabling parallelization), and to protect the privacy of the dataset.

**Remark 2.** Theorem 1 also holds for the simpler linear regression problem.

## 5 Experiments

We demonstrate the performance of COPML compared to conventional MPC baselines by examining two properties, accuracy and performance gain, in terms of the training time on the Amazon EC2 Cloud Platform.

### 5.1 Experiment setup

**Setup.** We train a logistic regression model for binary image classification on the CIFAR-10 [23] and GISETTE [18] datasets, whose size is $(m, d) = (9019, 3073)$ and $(6000, 5000)$, respectively. The dataset is distributed evenly across the clients. The clients initially secret share their individual datasets with the other clients.[5] Computations are carried out on Amazon EC2 `m3.xlarge` machine instances. We run the experiments in a WAN setting with an average bandwidth of $40Mbps$. Communication between clients is implemented using the `MPI4Py` [11] interface on `Python`.

**Implemented schemes.** We implement four schemes for performance evaluation. For COPML, we consider two set of key parameters $(K, T)$ to investigate the trade-off between parallelization and privacy. For the baselines, we apply two conventional MPC protocols (based on [4] and [3]) to our multi-client problem setting.[6]

1. **COPML.** In COPML, MPC is utilized to enable secure encoding and decoding for Lagrange coding. The gradient computations are then carried out using the Lagrange encoded data. We determine $T$ (privacy threshold) and $K$ (amount of parallelization) in COPML as follows. Initially, we have from Theorem 1 that these parameters must satisfy $N \geq (2r+1)(K+T-1)+1$ for our framework. Next, we have considered both $r = 1$ and $r = 3$ for the degree of the polynomial approximation of the sigmoid function and observed that the degree one approximation achieves good accuracy, as we demonstrate later. Given our choice of $r = 1$, we then consider two setups:

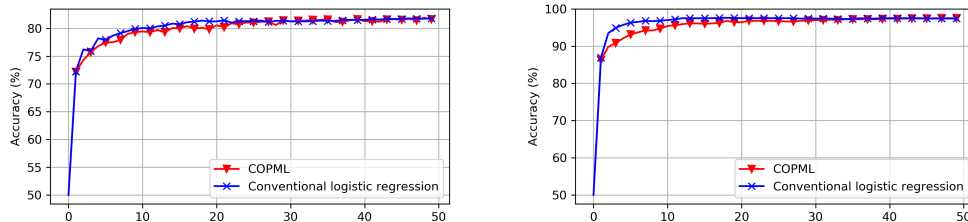

(a) CIFAR-10 dataset for binary classification be-tween *plain* and *car* images (using 9019 samples for the training set and 2000 samples for the test set).

(b) GISETTE dataset for binary classification be-tween digits 4 and 9 (using 6000 samples for the training set and 1000 samples for the test set).

Figure 4: Comparison of the accuracy of COPML (demonstrated for Case 2 and $N = 50$ clients) vs conventional logistic regression that uses the sigmoid function without quantization.

> **Case 1:** *(Maximum parallelization gain)* Allocate all resources to parallelization (fastest training), by letting $K = \lfloor \frac{N-1}{3} \rfloor$ and $T = 1$,
>
> **Case 2:** *(Equal parallelization and privacy gain)* Split resources almost equally between paral-lelization and privacy, i.e., $T = \lfloor \frac{N-3}{6} \rfloor$, $K = \lfloor \frac{N+2}{3} \rfloor - T$.

2. **Baseline protocols.** We implement two conventional MPC protocols (based on [4] and [3]). In a naive implementation of these protocols, each client would secret share its local dataset with the entire set of clients, and the gradient computations would be performed over the secret shared data whose size is as large as the entire dataset, which leads to a significant computational overhead. For a fair comparison with COPML, we speed up the baseline protocols by partitioning the clients into three groups, and assigning each group one third of the entire dataset. Hence, the total amount of data processed at each client is equal to one third of the size of the entire dataset, which significantly reduces the total training time while providing a privacy threshold of $T = \lfloor \frac{N-3}{6} \rfloor$, which is the same privacy threshold as Case 2 of COPML. The details of these implementations are presented in Appendix A.4.

In all schemes, we apply the MPC truncation protocol from Section 3 to carry out the multiplication with $\frac{\eta}{m}$ during model updates, by choosing $(k_1, k_2) = (21, 24)$ and $(22, 24)$ for the CIFAR-10 and GISETTE datasets, respectively.

## 5.2 Performance evaluation

**Training time.** In the first set of experiments, we measure the training time. Our results are demonstrated in Figure 3, which shows the comparison of COPML with the protocol from [3], as we have found it to be the faster of the two baselines. Figures 3(a) and 3(b) demonstrate that COPML provides substantial speedup over the MPC baseline, in particular, up to $8.6\times$ and $16.4\times$ with the CIFAR-10 and GISETTE datasets, respectively, while providing the same privacy threshold $T$. We observe that a higher amount of speedup is achieved as the dimension of the dataset becomes larger (CIFAR-10 vs. GISETTE datasets), suggesting COPML to be well-suited for data-intensive distributed training tasks where parallelization is essential.

To further investigate the gain of COPML, in Table 1 we present the breakdown of the total run-ning time with the CIFAR-10 dataset for $N = 50$ clients. We observe that COPML provides $K/3$ times speedup for the com-putation time of matrix multipli-

Table 1: Breakdown of the running time with $N = 50$ clients.

| Protocol | Comp. time (s) | Comm. time (s) | Enc/Dec time (s) | Total run time (s) |
|---|---|---|---|---|
| MPC using [BGW88] | 918 | 21142 | 324 | 22384 |
| MPC using [BH08] | 914 | 6812 | 189 | 7915 |
| COPML (Case 1) | 141 | 284 | 15 | 440 |
| COPML (Case 2) | 240 | 654 | 22 | 916 |

cation in (7), which is given in the first column. This is due to the fact that, in the baseline protocols, the size of the data processed at each client is one third of the entire dataset, while in COPML it is $(1/K)^{th}$ of the entire dataset. This reduces the computational overhead of each client while computing matrix multiplications. Moreover, COPML provides significant improvement in the communication, encoding, and decoding time. This is because the two baseline protocols require intensive communication and computation to carry out a degree reduction step for secure multi-plication (encoding and decoding for additional secret shares), which is detailed in Appendix A.3.

Table 2: Complexity summary of COPML.

| Communication | Computation | Encoding |
|---|---|---|
| $O(\frac{mdN}{K} + dNJ)$ | $O(\frac{md^2}{K})$ | $O(\frac{mdN(K+T)}{K} + dN(K+T)J)$ |

In contrast, COPML only requires secure addition and multiplication-by-a-constant operations for encoding and decoding. These operations require no communication. In addition, the communication, encoding, and decoding overheads of each client are also reduced due to the fact that the size of the data processed at each client is only $(1/K)^{th}$ of the entire dataset.

**Accuracy.** We finally examine the accuracy of COPML. Figures 4(a) and 4(b) demonstrate that COPML with degree one polynomial approximation provides comparable test accuracy to conventional logistic regression. For the CIFAR-10 dataset in Figure 4(a), the accuracy of COPML and conventional logistic regression are $80.45\%$ and $81.75\%$, respectively, in $50$ iterations. For the GISETTE dataset in Figure 4(b), the accuracy of COPML and conventional logistic regression have the same value of $97.5\%$ in $50$ iterations. Hence, COPML has comparable accuracy to conventional logistic regression while also being privacy preserving.

## 5.3 Complexity Analysis

In this section, we analyze the asymptotic complexity of each client in COPML with respect to the number of clients $N$, model dimension $d$, number of data points $m$, parallelization parameter $K$, privacy parameter $T$, and total number of iterations $J$. Client $i$'s communication cost can be broken to three parts: 1) sending the secret shares $[\widetilde{\mathbf{X}}_j]_i = [u(\alpha_j)]_i$ in (3) to client $j \in [N]$, 2) sending the secret shares $[\widetilde{\mathbf{w}}_j^{(t)}]_i = [v(\alpha_j)]_i$ in (4) to client $j \in [N]$ for $t \in \{0, \ldots, J-1\}$, and 3) sending the secret share of local computation $[f(\widetilde{\mathbf{X}}_i, \widetilde{\mathbf{w}}_i^{(t)})]_j$ in (7) to client $j \in [N]$ for $t \in \{0, \ldots, J-1\}$. The communication cost of the three parts are $O(\frac{mdN}{K})$, $O(dNJ)$, and $O(dNJ)$, respectively. Therefore, the overall communication cost of each client is $O(\frac{mdN}{K} + dNJ)$. Client $i$'s computation cost of encoding can be broken into two parts, encoding the dataset by using (3) and encoding the model by using (4). The encoded dataset $[\widetilde{\mathbf{X}}_j]_i = [u(\alpha_j)]_i$ from (3) is a weighted sum of $K+T$ matrices where each matrix belongs to $\mathbb{F}_p^{\frac{m}{K} \times d}$. As there are $N$ encoded dataset and each encoded dataset requires a computation cost of $O(\frac{md(K+T)}{K})$, the computation cost of encoding the dataset is $O(\frac{mdN(K+T)}{K})$ in total. Similarly, computation cost of encoding $[\widetilde{\mathbf{w}}_j^{(t)}]_i = [v(\alpha_j)]_i$ from (4) is $O(dN(K+T)J)$. Computation cost of client $i$ to compute $\widetilde{\mathbf{X}}_i^\top \widetilde{\mathbf{X}}_i$, the dominant part of local computation $f(\widetilde{\mathbf{X}}_i, \widetilde{\mathbf{w}}_i^{(t)})$ in (7), is $O(\frac{md^2}{K})$. We summarize the asymptotic complexity of each client in Table 2.

When we set $N = 3(K + T - 1) + 1$ and $K = O(N)$ (Case 2), increasing $N$ has two major impacts on the training time: 1) reducing the computation per client by choosing a larger $K$, 2) increasing the encoding time. In this case, as $m$ is typically much larger than other parameters, dominate terms in communication, computation, and encoding cost are $O(md)$, $O(md^2/N)$ and $O(mdN)$, respectively. For small datasets, i.e., when the computation load at each worker is very small, the gain from increasing the number of workers beyond a certain point may be minimal and system may saturate, as encoding may dominate the computation. This is the reason that a higher amount of speedup of training time is achieved as the dimension of the dataset becomes larger.

## 6 Conclusions

We considered a collaborative learning scenario in which multiple data-owners jointly train a logistic regression model without revealing their individual datasets to the other parties. To the best of our knowledge, even for the simple logistic regression, COPML is the first fully-decentralized training framework to scale beyond 3-4 parties while achieving information-theoretic privacy. Extending COPML to more complicated (deeper) models is a very interesting future direction. An MPC-friendly (i.e., polynomial) activation function is proposed in [28] which approximates the softmax and shows that the accuracy of the resulting models is very close to those trained using the original functions. We expect to achieve a similar performance gain even in those setups, since COPML can similarly be leveraged to efficiently parallelize the MPC computations.

## Broader Impact

Our framework has the societal benefit of protecting user privacy in collaborative machine learning applications, where multiple data-owners can jointly train machine learning models without revealing information about their individual datasets to the other parties, even if some parties collude with each other. Collaboration can significantly improve the accuracy of trained machine learning models, compared to training over individual datasets only. This is especially important in applications where data labelling is costly and can take a long time, such as data collected and labeled in medical fields. For instance, by using our framework, multiple medical institutions can collaborate to train a logistic regression model jointly, without revealing the privacy of their datasets to the other parties, which may contain sensitive patient healthcare records or genetic information. Our framework can scale to a significantly larger number of users compared to the benchmark protocols, and can be applied to any field in which the datasets contain sensitive information, such as healthcare records, financial transactions, or geolocation data. In such applications, protecting the privacy of sensitive information is critical and failure to do so can result in serious societal, ethical, and legal consequences. Our framework can provide both application developers and users with positive societal consequences, application developers can provide better user experience with better models as the volume and diversity of data will be increased greatly, and at the same time, users will have their sensitive information kept private. Another benefit of our framework is that it provides strong privacy guarantees that is independent from the computational power of the adversaries. Therefore, our framework keeps the sensitive user information safe even if adversaries gain quantum computing capabilities in the future.

A potential limitation of our framework is that our current training framework is bound to polynomial operations. In order to compute functions that are not polynomials, such as the sigmoid function, we utilize a polynomial approximation. This can pose a challenge in the future for applying our framework to deep neural network models, as the approximation error may add up at each layer. In such scenarios, one may need to develop additional techniques to better handle the non-linearities and approximation errors.

## Acknowledgement

This material is based upon work supported by Defense Advanced Research Projects Agency (DARPA) under Contract No. HR001117C0053, ARO award W911NF1810400, NSF grants CCF-1703575 and CCF-1763673, ONR Award No. N00014-16-1-2189, and research gifts from Intel and Facebook. The views, opinions, and/or findings expressed are those of the author(s) and should not be interpreted as representing the official views or policies of the Department of Defense or the U.S. Government.

## Footnotes

[1]COPML stands for collaborative privacy-preserving machine learning.

[2]Encoding of Lagrange coded computing is the same as a packed secret sharing [13].

[3]The random parameters can be generated by a crypto-service provider in an offline manner, or by using pseudo-random secret sharing [9].

[4]In fact, gathering only $T + 1$ secret shares is sufficient to recover $\widetilde{\mathbf{X}}_i$, due to the construction of Shamir's secret sharing [33]. Using this fact, one can speed up the execution by dividing the $N$ clients into subgroups of $T + 1$ and performing the encoding locally within each subgroup. We utilize this property in our experiments.

[5]This can be done offline as it is identical one-time operation for both MPC baselines and COPML.

[6]As described in the Section 1, there is no prior work at our scale (beyond 3-4 parties), hence we implement two baselines based on well-known MPC protocols which are also the first implementations at our scale.

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
