[Supplementary Material]

# A Supplementary Materials

## A.1 Details of the Quantization Phase

For quantizing its dataset $\mathbf{X}_j$, client $j \in [N]$ employs a scalar quantization function $\phi\left(Round(2^{l_x} \cdot \mathbf{X}_j)\right)$, where the rounding operation

$$Round(x) = \begin{cases} \lfloor x \rfloor & \text{if} \quad x - \lfloor x \rfloor < 0.5 \\ \lfloor x \rfloor + 1 & \text{otherwise} \end{cases} \tag{13}$$

is applied element-wise to the elements $x$ of matrix $\mathbf{X}_j$ and $l_x$ is an integer parameter to control the quantization loss. $\lfloor x \rfloor$ is the largest integer less than or equal to $x$, and function $\phi : \mathbb{Z} \to \mathbb{F}_p$ is a mapping defined to represent a negative integer in the finite field by using two's complement representation,

$$\phi(x) = \begin{cases} x & \text{if } x \geq 0 \\ p + x & \text{if } x < 0 \end{cases} \tag{14}$$

To avoid a wrap-around which may lead to an overflow error, prime $p$ should be large enough, $p \geq 2^{l_x+1} \max\{|x|\} + 1$. Its value also depends on the bitwidth of the machine as well as the dimension of the dataset. For example, in a $64$-bit implementation with the CIFAR-10 dataset whose dimension is $d = 3072$, we select $p = 2^{26} - 5$, which is the largest prime needed to avoid an overflow on intermediate multiplications. In particular, in order to speed up the running time of matrix-matrix multiplication, we do a modular operation after the inner product of vectors instead of doing a modular operation per product of each element. To avoid an overflow on this, $p$ should be smaller than a threshold given by $d(p-1)^2 \leq 2^{64} - 1$. For ease of exposition, throughout the paper, $\mathbf{X} = [\mathbf{X}_1^\top, \ldots, \mathbf{X}_N^\top]^\top$ refers to the quantized dataset.

## A.2 Proof of Theorem 1

First, we show that the minimum number of clients needed for our decoding operation to be successful, i.e., the recovery threshold of COPML, is equal to $(2r+1)(K+T-1)+1$. To do so, we demonstrate in the following that the decoding process will be successful as long as $N \geq (2r+1)(K+T-1)+1$. As described in Section 3, given the polynomial approximation of the sigmoid function in (5), the degree of $h(z)$ in (8) is at most $(2r+1)(K+T-1)$. The decoding process uses the computations from the clients as evaluation points $h(\alpha_i)$ to interpolate the polynomial $h(z)$. If at least $deg(h(z))+1$ evaluation results of $h(\alpha_i)$ are available, then, all of the coefficients of $h(z)$ can be evaluated. After $h(z)$ is recovered, the sub-gradient $\mathbf{X}_i^\top \hat{g}(\mathbf{X}_i \times \mathbf{w}^{(t)})$ can be decoded by computing $h(\beta_i)$ for $i \in [K]$, from which the gradient $\mathbf{X}^\top \hat{g}(\mathbf{X} \times \mathbf{w}^{(t)})$ from (11) can be computed. Hence, the recovery threshold of COPML is $(2r+1)(K+T-1)+1$, as long as $N \geq (2r+1)(K+T-1)+1$, the protocol can correctly decode the gradient using the local evaluations of the clients, and the decoding process will be successful. Since the decoding operations are performed using a secure MPC protocol, throughout the decoding process, the clients only learn a secret share of the gradient and not its actual value. Next, we consider the update equation in (6) and prove its convergence to $\mathbf{w}^*$. As described in Section 3, after decoding the gradient, the clients carry out a secure truncation protocol to multiply $\mathbf{X}^\top(\hat{g}(\mathbf{X} \times \mathbf{w}^{(t)}) - \mathbf{y})$ with parameter $\frac{\eta}{m}$ to update the model as in (6). The update equation from (6) can then be represented by

$$\mathbf{w}^{(t+1)} = \mathbf{w}^{(t)} - \eta\left(\frac{1}{m}\mathbf{X}^\top(\hat{g}(\mathbf{X} \times \mathbf{w}^{(t)}) - \mathbf{y}) + \mathbf{n}^{(t)}\right). \tag{15}$$

$$= \mathbf{w}^{(t)} - \eta\mathbf{p}^{(t)} \tag{16}$$

where $\mathbf{n}^{(t)}$ represents the quantization noise introduced by the secure multi-party truncation protocol [6], and $\mathbf{p}^{(t)} \triangleq \frac{1}{m}\mathbf{X}^\top(\hat{g}(\mathbf{X} \times \mathbf{w}^{(t)}) - \mathbf{y}) + \mathbf{n}^{(t)}$. From [6], $\mathbf{n}^{(t)}$ has zero mean and bounded variance, i.e., $\mathbb{E}_{\mathbf{n}^{(t)}}[\mathbf{n}^{(t)}] = 0$ and $\mathbb{E}_{\mathbf{n}^{(t)}}\left[\|\mathbf{n}^{(t)}\|_2^2\right] \leq \frac{d2^{2(k_1-1)}}{m^2} \triangleq \sigma^2$ where $\|\cdot\|_2$ is the $l_2$ norm and $k_1$ is the truncation parameter described in Section 3.

Next, we show that $\mathbf{p}^{(t)}$ is an unbiased estimator of the true gradient, $\nabla C(\mathbf{w}^{(t)}) = \frac{1}{m}\mathbf{X}^\top(g(\mathbf{X} \times \mathbf{w}^{(t)}) - \mathbf{y})$, and its variance is bounded by $\sigma^2$ with sufficiently large $r$. From $\mathbb{E}_{\mathbf{n}^{(t)}}[\mathbf{n}^{(t)}] = 0$, we obtain

$$\mathbb{E}_{\mathbf{n}^{(t)}}[\mathbf{p}^{(t)}] - \nabla C(\mathbf{w}^{(t)}) = \frac{1}{m}\mathbf{X}^\top\left(\hat{g}(\mathbf{X} \times \mathbf{w}^{(t)}) - g(\mathbf{X} \times \mathbf{w}^{(t)})\right). \tag{17}$$

From the Weierstrass approximation theorem [5], for any $\epsilon > 0$, there exists a polynomial that approximates the sigmoid arbitrarily well, i.e., $|\hat{g}(x) - g(x)| \leq \epsilon$ for all $x$ in the constrained interval. Hence, as there exists a polynomial making the norm of (17) arbitrarily small, $\mathbb{E}_{\mathbf{n}^{(t)}}[\mathbf{p}^{(t)}] = \nabla C(\mathbf{w}^{(t)})$ and $\mathbb{E}_{\mathbf{n}^{(t)}}\left[\|\mathbf{p}^{(t)} - \mathbb{E}_{\mathbf{n}^{(t)}}[\mathbf{p}^{(t)}]\|_2^2\right] = \mathbb{E}_{\mathbf{n}^{(t)}}\left[\|\mathbf{n}^{(t)}\|_2^2\right] \leq \sigma^2$.

Next, we consider the update equation in (16) and prove its convergence to $\mathbf{w}^*$. From the $L$-Lipschitz continuity of $\nabla C(\mathbf{w})$ (Theorem 2.1.5 of [29]), we have

$$C(\mathbf{w}^{(t+1)}) \leq C(\mathbf{w}^{(t)}) + \langle \nabla C(\mathbf{w}^{(t)}), \mathbf{w}^{(t+1)} - \mathbf{w}^{(t)} \rangle + \frac{L}{2}\| \mathbf{w}^{(t+1)} - \mathbf{w}^{(t)} \|^2$$

$$\leq C(\mathbf{w}^{(t)}) - \eta \langle \nabla C(\mathbf{w}^{(t)}), \mathbf{p}^{(t)} \rangle + \frac{L\eta^2}{2}\|\mathbf{p}^{(t)}\|^2, \tag{18}$$

where $\langle \cdot, \cdot \rangle$ is the inner product. For a cross entropy loss $C(\mathbf{w})$, the Lipschitz constant $L$ is equal to the largest eigenvalue of the Hessian $\nabla^2 C(\mathbf{w})$ for all $\mathbf{w}$, and is given by $L = \frac{1}{4}\|\mathbf{X}\|_2^2$. By taking the expectation with respect to the quantization noise $\mathbf{n}^{(t)}$ on both sides in (18), we have

$$\mathbb{E}_{\mathbf{n}^{(t)}}\left[C(\mathbf{w}^{(t+1)})\right] \leq C(\mathbf{w}^{(t)}) - \eta\| \nabla C(\mathbf{w}^{(t)})\|^2 + \frac{L\eta^2}{2}\left(\| \nabla C(\mathbf{w}^{(t)})\|^2 + \sigma^2\right) \tag{19}$$

$$\leq C(\mathbf{w}^{(t)}) - \eta\left(1 - \frac{L\eta}{2}\right)\| \nabla C(\mathbf{w}^{(t)})\|^2 + \frac{L\eta^2\sigma^2}{2}$$

$$\leq C(\mathbf{w}^{(t)}) - \frac{\eta}{2}\| \nabla C(\mathbf{w}^{(t)})\|^2 + \frac{\eta\sigma^2}{2} \tag{20}$$

$$\leq C(\mathbf{w}^*) + \langle \nabla C(\mathbf{w}^{(t)}), \mathbf{w}^{(t)} - \mathbf{w}^* \rangle - \frac{\eta}{2}\| \nabla C(\mathbf{w}^{(t)})\|^2 + \frac{\eta\sigma^2}{2} \tag{21}$$

$$\leq C(\mathbf{w}^*) + \langle \mathbb{E}_{\mathbf{n}^{(t)}}[\mathbf{p}^{(t)}], \mathbf{w}^{(t)} - \mathbf{w}^* \rangle - \frac{\eta}{2}\mathbb{E}_{\mathbf{n}^{(t)}}\|\mathbf{p}^{(t)}\|^2 + \eta\sigma^2 \tag{22}$$

$$= C(\mathbf{w}^*) + \eta\sigma^2 + \mathbb{E}_{\mathbf{n}^{(t)}}\left[\langle \mathbf{p}^{(t)}, \mathbf{w}^{(t)} - \mathbf{w}^* \rangle - \frac{\eta}{2}\|\mathbf{p}^{(t)}\|^2\right]$$

$$= C(\mathbf{w}^*) + \eta\sigma^2 + \frac{1}{2\eta}\left(\| \mathbf{w}^{(t)} - \mathbf{w}^* \|^2 - \mathbb{E}_{\mathbf{n}^{(t)}}\| \mathbf{w}^{(t+1)} - \mathbf{w}^*\|^2\right) \tag{23}$$

where (19) and (22) hold since $\mathbb{E}_{\mathbf{n}^{(t)}}[\mathbf{p}^{(t)}] = \nabla C(\mathbf{w}^{(t)})$ and $\mathbb{E}_{\mathbf{n}^{(t)}}\left[\|\mathbf{p}^{(t)} - \nabla C(\mathbf{w}^{(t)})\|_2^2\right] \leq \sigma^2$, (20) follows from $L\eta \leq 1$, (21) follows from the convexity of $C$, and (23) follows from $\mathbf{p}^{(t)} = -\frac{1}{\eta}(\mathbf{w}^{(t+1)} - \mathbf{w}^{(t)})$.

By taking the expectation on both sides in (23) with respect to the joint distribution of all random variables $\mathbf{n}^{(0)}, \ldots, \mathbf{n}^{(J-1)}$ where $J$ denotes the total number of iterations, we have

$$\mathbb{E}\left[C(\mathbf{w}^{(t+1)})\right] - C(\mathbf{w}^*) \leq \frac{1}{2\eta}\left(\mathbb{E}\| \mathbf{w}^{(t)} - \mathbf{w}^* \|^2 - \mathbb{E}\| \mathbf{w}^{(t+1)} - \mathbf{w}^*)\|^2\right) + \eta\sigma^2. \tag{24}$$

Summing both sides of the inequality in (24) for $t = 0, \ldots, J - 1$, we find that,

$$\sum_{t=0}^{J-1}\left(\mathbb{E}\left[C(\mathbf{w}^{(t+1)})\right] - C(\mathbf{w}^*)\right) \leq \frac{\| \mathbf{w}^{(0)} - \mathbf{w}^* \|^2}{2\eta} + J\eta\sigma^2.$$

Finally, since $C$ is convex, we observe that,

$$\mathbb{E}\left[C\left(\frac{1}{J}\sum_{t=0}^{J}\mathbf{w}^{(t)}\right)\right] - C(\mathbf{w}^*) \leq \frac{1}{J}\sum_{t=0}^{J-1}\left(\mathbb{E}\left[C(\mathbf{w}^{(t+1)})\right] - C(\mathbf{w}^*)\right)$$

$$\leq \frac{\| \mathbf{w}^{(0)} - \mathbf{w}^* \|^2}{2\eta J} + \eta\sigma^2$$

which completes the proof of convergence.

### A.3 Details of the Multi-Party Computation (MPC) Implementation

We consider two well-known MPC protocols, the notable BGW protocol from [4], and the more recent, efficient MPC protocol from [3, 12]. Both protocols allow the computation of any polynomial

function in a privacy-preserving manner by untrusted parties. Computations are carried out over the secret shares, and at the end, parties only learn a secret share of the actual result. Any collusions between up to $T = \lfloor \frac{N-1}{2} \rfloor$ out of $N$ parties do not reveal information (in an information-theoretic sense) about the input variables. The latter protocol is more efficient in terms of the communication cost between the parties, which scales linearly with respect to the number of parties, whereas for the former protocol this cost is quadratic. As a trade-off, it requires a considerable amount of offline computations and higher storage cost for creating and secret sharing the random variables used in the protocol.

For creating secret shares, we utilize Shamir's $T$-out-of-$N$ secret sharing [33]. This scheme embeds a secret $a$ in a degree $T$ polynomial $h(\xi) = a + \xi v_1, \dots, \xi^T v_T$ where $v_i$, $i \in [T]$ are uniformly random variables. Client $i \in [N]$ then receives a secret share of $a$, denoted by $h(i) = [a]_i$. This keeps $a$ private against any collusions between up to any $T$ parties. The specific computations are then carried out as follows.

**Addition.** In order to perform a secure addition $a + b$, clients locally add their secret shares $[a]_i + [b]_i$. The resulting value is a secret share of the original summation $a + b$. This step requires no communication.

**Multiplication-by-a-constant.** For performing a secure multiplication $ac$ where $c$ is a publicly-known constant, clients locally multiply their secret share $[a]_i$ with $c$. The resulting value is a secret share of the desired multiplication $ac$. This step requires no communication.

**Multiplication.** For performing a secure multiplication $ab$, the two protocols differ in their execution. In the BGW protocol, each client initially multiplies its secret shares $[a]_i$, $[b]_i$ locally to obtain $[a]_i[b]_i$. The clients will then be holding a secret share of $ab$, however, the corresponding polynomial now has degree $2T$. This may in turn cause the degree of the polynomial to increase excessively as more multiplication operations are evaluated. To alleviate this problem, in the next phase, clients carry out a degree reduction step to create new shares corresponding to a polynomial of degree $T$. The communication overhead of this protocol is $O(N^2)$.

The protocol from [3], on the other hand, leverages offline computations to speed up the communication phase. In particular, a random variable $\rho$ is created offline and secret shared with the clients twice using two random polynomials with degrees $T$ and $2T$, respectively. The secret shares corresponding to the degree $T$ polynomial are denoted by $[\rho]_{T,i}$, whereas the secret shares for the degree $2T$ polynomial are denoted by $[\rho]_{2T,i}$ for clients $i \in [N]$. In the online phase, client $i \in [N]$ locally computes the multiplication $[a]_i[b]_i$, after which each client will be holding a secret share of the multiplication $ab$. The resulting polynomial has degree $2T$. Then, each client locally computes $[a]_i[b]_i - [\rho]_{2T,i}$, which corresponds to a secret share of $ab - \rho$ embedded in a degree $2T$ polynomial. Clients then broadcast their individual computations to others, after which each client computes $ab - \rho$. Note that the privacy of the computation $ab$ is still protected since clients do not know the actual value of $ab$, but instead its masked version $ab - \rho$. Then, each client locally computes $ab - \rho + [\rho]_{T,i}$. As a result, variable $\rho$ cancels out, and clients obtain a secret share of the multiplication $ab$ embedded in a degree $T$ polynomial. This protocol requires only $O(N)$ broadcasts and therefore is more efficient than the previous algorithm. On the other hand, it requires an offline computation phase and higher storage overhead. For the details, we refer to [3, 2].

**Remark 3.** The secure MPC computations during the encoding, decoding, and model update phases of COPML only use addition and multiplication-by-a-constant operations, instead of the expensive multiplication operation, as $\{\alpha_i\}_{i \in [N]}$ and $\{\beta_k\}_{k \in [K+T]}$ are publicly known constants for all clients.

### A.4 Details of the Optimized Baseline Protocols

In a naive implementation of our multi-client problem setting, both baseline protocols would utilize Shamir's secret sharing scheme where the quantized dataset $\mathbf{X} = [\mathbf{X}_1^\top, \dots, \mathbf{X}_N^\top]^\top$ is secret shared with $N$ clients. To do so, both baselines would follow the same secret sharing process as in COPML, where client $j \in [N]$ creates a degree $T$ random polynomial $h_j(z) = \mathbf{X}_j + z\mathbf{R}_{j1} + \dots + z^T\mathbf{R}_{jT}$ where $\mathbf{R}_{ji}$ for $i \in [T]$ are i.i.d. uniformly distributed random matrices while selecting $T = \lfloor \frac{N-1}{2} \rfloor$. By selecting $N$ distinct evaluation points $\lambda_1, \dots, \lambda_N$ from $\mathbb{F}_p$, client $j$ would generate and send $[\mathbf{X}_j]_i = h_j(\lambda_i)$ to client $i \in [N]$. As a result, client $i \in [N]$ would be assigned a secret share of the entire dataset $\mathbf{X}$, i.e, $[\mathbf{X}]_i = \left[[\mathbf{X}_1]_i^\top, \dots, [\mathbf{X}_N]_i^\top\right]^\top$. Client $i$ would also obtain a secret share of the

labels, $[\mathbf{y}]_i$, and a secret share of the initial model, $[\mathbf{w}^{(0)}]_i$, where $\mathbf{y} = [\mathbf{y}_1^\top, \ldots, \mathbf{y}_N^\top]^\top$ and $\mathbf{w}^{(0)}$ is a randomly initialized model. Then, the clients would compute the gradient and update the model from (7) within a secure MPC protocol. This guarantees privacy against $\lfloor \frac{N-1}{2} \rfloor$ colluding workers, but requires a computation load at each worker that is as large as processing the whole dataset at a single worker, leading to slow training.

Hence, in order to provide a fair comparison with COPML, we optimize (speed up) the baseline protocols by partitioning the clients into subgroups of size $2T + 1$. Clients communicate a secret share of their own datasets with the other clients in the same subgroup, instead of secret sharing it with the entire set of clients. Each client in subgroup $i$ receives a secret share of a partitioned dataset $\mathbf{X}_i \in \mathbb{F}_p^{\frac{m}{G} \times d}$ where $\mathbf{X} = [\mathbf{X}_1^\top \cdots \mathbf{X}_G^\top]^\top$ and $G$ is the number of subgroups. In other words, client $j$ in subgroup $i$ obtains a secret share $[\mathbf{X}_i]_j$. Then, subgroup $i \in [G]$ computes the sub-gradient over the partitioned dataset, $\mathbf{X}_i$, within a secure MPC protocol. To provide the same privacy threshold $T = \lfloor \frac{N-3}{6} \rfloor$ as Case 2 of COPML in Section 5, we set $G = 3$. This significantly reduces the total training time of the two baseline protocols (compared to the naive MPC implementation where the computation load at each client would be as high as training centrally), as the total amount of data processed at each client is equal to one third of the size of the entire dataset $\mathbf{X}$.

## A.5 Algorithms

The overall procedure of COPML protocol is given in Algorithm 1.

---
**Algorithm 1** COPML
---
**input** Dataset $(\mathbf{X}, \mathbf{y}) = ((\mathbf{X}_1, \mathbf{y}_1), \ldots, (\mathbf{X}_N, \mathbf{y}_N))$ distributed over $N$ clients.
**output** Model parameters $\mathbf{w}^{(J)}$.

1: **for** client $j = 1, \ldots, N$ **do**
2:     Secret share the individual dataset $(\mathbf{X}_j, \mathbf{y}_j)$ with clients $i \in [N]$.
3: **end for**
4: Within a secure MPC protocol, initialize the model $\mathbf{w}^{(0)}$ randomly and secret share with clients $i \in [N]$.
    *// Client $i$ receives a secret share $[\mathbf{w}^{(0)}]_i$ of $\mathbf{w}^{(0)}$.*
5: Encode the dataset within a secure MPC protocol, using the secret shares $[\mathbf{X}_j]_i$ for $j \in [N], i \in [N]$.
    *// After this step, client $i$ holds a secret share $[\widetilde{\mathbf{X}}_j]_i$ of each encoded dataset $\widetilde{\mathbf{X}}_j$ for $j \in [N]$.*
6: **for** client $i = 1, \ldots, N$ **do**
7:     Gather the secret shares $[\widetilde{\mathbf{X}}_i]_j$ from clients $j \in [N]$.
8:     Recover the encoded dataset $\widetilde{\mathbf{X}}_i$ from the secret shares $\{[\widetilde{\mathbf{X}}_i]_j\}_{j \in [N]}$.
        *// At the end of this step, client $i$ obtains the encoded dataset $\widetilde{\mathbf{X}}_i$.*
9: **end for**
10: Compute $\mathbf{X}^T \mathbf{y}$ within a secure MPC protocol using the secret shares $[\mathbf{X}_j]_i$ and $[\mathbf{y}_j]_i$ for $j \in [N], i \in [N]$.
    *// At the end of this step, client $i$ holds a secret share $[\mathbf{X}^T \mathbf{y}]_i$ of $\mathbf{X}^T \mathbf{y}$.*
11: **for** iteration $t = 0, \ldots, J - 1$ **do**
12:     Encode the model $\mathbf{w}^{(t)}$ in a secure MPC protocol using the secret shares $[\mathbf{w}^{(t)}]_i$.
        *// After this step, client $i$ holds a secret share $[\widetilde{\mathbf{w}}_j^{(t)}]_i$ of the encoded model $\widetilde{\mathbf{w}}_j^{(t)}$ for $j \in [N]$.*
13:     **for** client $i = 1, \ldots, N$ **do**
14:         Gather the secret shares $[\widetilde{\mathbf{w}}_i^{(t)}]_j$ from clients $j \in [N]$.
15:         Recover the encoded model $\widetilde{\mathbf{w}}_i^{(t)}$ from the secret shares $\{[\widetilde{\mathbf{w}}_i^{(t)}]_j\}_{j \in [N]}$.
            *// At the end of this step, client $i$ obtains the encoded model $\widetilde{\mathbf{w}}_i^{(t)}$.*
16:         Locally compute $f(\widetilde{\mathbf{X}}_i, \widetilde{\mathbf{w}}_i^{(t)})$ from (7) and secret share the result with clients $j \in [N]$.
            *// Client $i$ sends a secret share $[f(\widetilde{\mathbf{X}}_i, \widetilde{\mathbf{w}}_i^{(t)})]_j$ of $f(\widetilde{\mathbf{X}}_i, \widetilde{\mathbf{w}}_i^{(t)})$ to client $j$.*
17:     **end for**
18:     **for** client $i = 1, \ldots, N$ **do**
19:         Locally computes $[f(\mathbf{X}_k, \mathbf{w}^{(t)})]_i$ for $k \in [K]$ from (10).
            *// After this step, client $i$ knows a secret share $[f(\mathbf{X}_k, \mathbf{w}^{(t)})]_i$ of $f(\mathbf{X}_k, \mathbf{w}^{(t)})$ for $k \in [K]$.*
20:         Locally aggregate the secret shares $\{[f(\mathbf{X}_k, \mathbf{w}^{(t)})]_i\}_{k \in \mathcal{K}}$ to compute $[\mathbf{X}^T \hat{g}(\mathbf{X} \times \mathbf{w}^{(t)})]_i$ $\triangleq$ $\sum_{k \in [K]} [f(\mathbf{X}_k, \mathbf{w}^{(t)})]_i$.
            *// At the end of this step, client $i$ now has a secret share $[\mathbf{X}^T \hat{g}(\mathbf{X} \times \mathbf{w}^{(t)})]_i$ of $\mathbf{X}^T \hat{g}(\mathbf{X} \times \mathbf{w}^{(t)}) = \sum_{k \in [K]} f(\mathbf{X}_k, \mathbf{w}^{(t)})$.*
21:         Locally compute $[\mathbf{X}^\top (\hat{g}(\mathbf{X} \times \mathbf{w}^{(t)}) - \mathbf{y})]_i \triangleq [\mathbf{X}^T \hat{g}(\mathbf{X} \times \mathbf{w}^{(t)})]_i - [\mathbf{X}^T \mathbf{y}]_i$.
            *// Each client now has a secret share $[\mathbf{X}^\top (\hat{g}(\mathbf{X} \times \mathbf{w}^{(t)}) - \mathbf{y})]_i$ of $\mathbf{X}^\top (\hat{g}(\mathbf{X} \times \mathbf{w}^{(t)}) - \mathbf{y})$.*
22:     **end for**
23:     Update the model according to (6) within a secure MPC protocol using the secret shares $[\mathbf{X}^\top (\hat{g}(\mathbf{X} \times \mathbf{w}^{(t)}) - \mathbf{y})]_i$ and $[\mathbf{w}^{(t)}]_i$ for $i \in [N]$, and by carrying out the secure truncation operation.
        *// At the end of this step, client $i$ holds a secret share of the updated model $[\mathbf{w}^{(t+1)}]_i$.*
        *// Secure truncation is carried out jointly as it requires communication between the clients.*
24: **end for**
25: **for** client $j = 1, \ldots, N$ **do**
26:     Collect the secret shares $[\mathbf{w}^{(J)}]_i$ from clients $i \in [N]$ and recover the final model $\mathbf{w}^{(J)}$.
27: **end for**
---