[Reviews · NeurIPS 2020]

Review 1

Summary and Contributions: The paper presents an privacy-preserving protocol for learning a logistic regression model on data distributed across multiple data holders. The protocol utilizes secret sharing and Lagrange encodings, and provides informational theoretic privacy against semi-honest adversaries. The protocol tolerates a customizable number of colluding parties.

Strengths: The ideas presented in the paper are clearly justified (proofs in the Appendix) and seem to be correct, although I did not check all details. Key contributions of the protocol are the capability to distribute the computational cost across multiple parties, and the reduction in communication cost due to the use of Lagrange encodings instead of more communication-heavy MPC computations. The authors implemented two baseline protocols to compare against, and as far as I can tell the comparisons are meaningful and fair. The results are particularly powerful due to flexibility in the protocol parameters (T, K, N, r).

Weaknesses: My biggest concerns is the the linear approximation to sigmoid. Does this result in essentially learning a linear regression model instead of logistic regression? If so, how does this scale to larger r? What happens to the communication cost? I also feel like the empirical evaluation is somewhat lacking: - The only scenario evaluated is one with 50 data providers. How does this scale to more or less? - What exactly is the communication size? It seems that the communication time is dominating in all protocols, so it would be important to provide more details on that. - For the baseline protocols, was a similar linear approximation of sigmoid used?

Correctness: The claims and methods seem correct and justified, but the empirical evaluation is too narrow to give a clear picture of how well this method works beyond the small examples presented.

Clarity: The paper is clear and easy to understand for people with a background in cryptography.

Relation to Prior Work: Prior work is certainly discussed, but I'm not sure if the main differences to prior work are made sufficiently clear for a broader audience to understand.

Reproducibility: Yes

Additional Feedback: - The graphs in Figure 4 are kind of hard to read because they are so small and the graphs are all clustered together. Maybe a table would be better to represent these results. - There is a typo in the caption of Figure 4a: it says "plain" instead of "plane".


Review 2

Summary and Contributions: This paper proposed a fully-decentralized training framework that achieves scalability and privacy-protection. The key idea is to securely encode the individual datasets to distribute the computation load effectively across many parties and to perform the training computations as well as the model updates in a distributed manner on the securely encoded data. Convergence analysis and experimentation are provided.

Strengths: This paper is generally well-organised and well-written.

Weaknesses: 1. The major concern of the reviewer is on the novelty. All the adopted techniques are well-known, this paper just combines previous techniques into one framework. 2. FL with MPC is especially susceptible to poisoning attacks as the individual updates cannot be inspected. 3. Lack of theoretic support on the statement "Our framework can provide strong privacy guarantees against colluding parties with unbounded computational power". 4. Lack of comparison with the most recent MPC protocols, all the compared protocols were proposed 10 years ago. 5. Figure 3 is weird, training time of CodedPrivateML even decreases with the increasing number of of clients N, which seems counterfactual, any explanation here? 6. Implementation on simple logistic regression model cannot fully validate the scalability of CodedPrivateML. It is expected to see how CodedPrivateML performs on more complex DNN models.

Correctness: Probably correct

Clarity: yes

Relation to Prior Work: Lack of comparison with the most recent MPC protocols, all the compared protocols were proposed 10 years ago.

Reproducibility: Yes

Additional Feedback:


Review 3

Summary and Contributions: The authors present distributed privacy-preserving logistic regression training using packed Shamir secret sharing. The degree of distribution and number of corrupted parties can be set independently, thus allowing exploring a space of computation cost and security considerations. Benchmarks are given in comparison to using plain Shamir secret sharing.

Strengths: Correctness and security are evident. The application of packed Shamir secret sharing to distribute the training computation is interesting.

Weaknesses: There isn't much novelty in terms of the components. Post rebuttal: I'm satisfied with the rebuttal, and I agree in particular with the authors' point that there isn't much recent work to compare with.

Correctness: I cannot find any faults.

Clarity: The paper is very well written in terms of grammar style, but there is potential for improvements in terms of clarity: Section 3 spells out Lagrange interpolation several times. I think the paper would benefit from a general introduction of the Secret sharing scheme followed by a more abstract description of the protocol.

Relation to Prior Work: The authors attribute their main technique to Yu et al. (AISTATS '19). However, it was first proposed by Frankling and Yung (STOC '92), and subsequent works have established the name packed secret sharing (Damgard et al., Eurocrypt '10). I find the term Lagrange coded computing confusing.

Reproducibility: Yes

Additional Feedback: Have the authors considered using pseudo-random secret sharing (Damgard et al., TCC '06) to generate the random parameters Z_k instead of a "crypto-service provider)?


Review 4

Summary and Contributions: The paper aims at proposing a fast encryption based method for training a logistic regression model privately, on data from multiple parties.

Strengths: The problem of training ML models securely and privately is a very relevant problem, and I commend the authors for addressing it and trying to provide speedup.

Weaknesses: The paper claims 16X speedup over a baseline from 1988, and 8X over a baseline from 2008. I wonder if there are any more recent works to compare with? I know that for deep learning there is constant improvement in the speed of encryption based methods, however I am not entirely familiar with the existing work for algorithms with machine learning. For example you could maybe compare to [25] and [26] in the paper.

Correctness: The paper seems solid in terms of experimentation and reasoning. However I am not an expert in encryption.

Clarity: yes

Relation to Prior Work: The paper has cited related work, but does not very well explain the difference between their contributions and prior work.

Reproducibility: Yes

Additional Feedback: I would like to see comparison to more recent work, and also I would like to see a break down of memory consumption and communication costs, since communication is also a bottleneck in these systems. ___________ After rebuttal: The rebuttal addressed my concern, and I have updated my score accordingly.

[Author Response · NeurIPS 2020]

We thank all reviewers for their helpful comments, and provide our response below (reviewers' comments are italicized).

**Reviewer 1.** 1) [*Polynomial approximation to sigmoid*] We in fact experimented with both $r = 1$ and $r = 3$ for the degree of the polynomial approximation to sigmoid, but observed that $r = 1$ achieved comparable test accuracy to conventional logistic regression, hence only reported $r = 1$. CodedPrivateML can be applied to any $r$ satisfying $N \geq (2r + 1)(K + T - 1) + 1$, where $N, K, T$ represent the number of clients, parallelization parameter, privacy parameter, respectively. The same approximation was also used for the baselines, hence using a larger $r$ would have the same impact on all protocols in terms of accuracy. In terms of total training time, the performance gain remains the same because the training time of both CodedPrivateML and baseline protocols increases linearly as $r$ increases.

2) [*Empirical evaluation with only 50 clients*] We would have liked to go beyond 50 clients, in fact our gains would be even better, but we are limited by the budget cost of running our experiments on the Amazon EC2 cloud. Currently our range is between 10-50 clients, which, compared to the state-of-the-art, already corresponds to a $10\times$ increase.

3) [*Communication cost and size*] Our communication cost per client is $O(mdN/K + dNJ)$ where $J$ denotes the number of iterations. When $N = 50$, the communication size per client is 126MB with the CIFAR-10 dataset. We note that increasing $N$ has two major impacts on the training time: 1) reducing the computation time by choosing a larger $K = O(N)$, and 2) increasing the communication time. Hence, when $N$ is small ($\sim 10$), the computation time dominates the total training time. For the baseline protocols, both communication and computation time increases as $N$ increases, and when $N = 50$, the communication size per client is 900MB. We will add this analysis in our revised version. We will also apply the changes suggested by the reviewer to improve the overall presentation.

**Reviewer 2.** We would like to clarify/correct a few key points in reviewer's comments, with the hope that this will help highlight the key contributions of our paper.

1) [*Novelty*] It is true that much of the privacy-preserving machine learning (PPML) literature is based on well-known homomorphic encryption or secure MPC primitives. Our contribution, on the other hand, is building the first PPML framework that can scale to a significantly larger number of clients than state-of-the-art PPML approaches (i.e., beyond 3-4 clients) with strong (information-theoretic or statistical) privacy guarantees. This is the first PPML approach that reduces the computation load per client as the number of clients increases, which we hope will open up further research.

2) [*Susceptibility to poisoning*] Our focus is on semi-honest (passive) adversaries (as stated in Section 1), which precludes poisoning attacks. PPML in general is susceptible to poisoning attacks, and this is also true for all MPC-based PPML schemes that we compare with (and build upon). Poisoning attacks is certainly an interesting future direction.

3) [*Privacy against unbounded adversaries*] We would like to emphasize that information-theoretic and statistical privacy are synonyms for "privacy against computationally unbounded adversaries". As discussed after Theorem 1, our privacy guarantee follows from the fact that all building blocks of our algorithm guarantees either information-theoretic privacy or statistical privacy of the individual datasets against any collusions between up to $T$ clients.

4) [*Lack of comparison with the most recent MPC protocols*]: Please see our response to Comment 1 from Reviewer 4.

5) [*Figure 3 - training time decreases*] This is in fact the main contribution of our paper. Our encoding procedure decreases the computation load per client (for training) as the number of clients increases, hence decreasing the overall training time as observed in Figure 3.b. Figure 3.a demonstrates a smaller dataset, in which case the encoding time starts to dominate over the benefits gained from encoding beyond 20-30 clients, and the training time starts to increase.

6) [*Extension to DNN*] This paper focuses on simpler models, and is a first step towards realizing scalable PPML for DNNs. Please note that, even in this domain, the problem is already very challenging and an active area of research.

**Reviewer 3.** 1) [*Packed secret sharing*] Thank you for raising this. We agree with the reviewer that the encoding of Lagrange coded computing is the same as packed secret sharing while [36] proves its optimality in terms of *recovery threshold* to compute an arbitrary multivariate polynomial. Recovery threshold is defined as the minimum number of computation results that the protocol needs to wait to guarantee decodability. We will add these points in our final paper.

2) [*Pseudo-random secret sharing*] Thank you for this suggestion, we agree with utilizing pseudo-random secret sharing to generate the random parameters instead of a crypto-service provider, and will incorporate this in our final paper.

**Reviewer 4.** 1) [*Comparison to more recent work*] There are in fact more recent MPC-based privacy-preserving machine learning (PPML) protocols, but we are not aware of any work that goes beyond 3-4 parties. Our work is the first PPML solution that goes substantially beyond that. As there was no prior work at our scale, we implemented two baselines based on well-known MPC protocols which are also the first implementations at that scale. Following the reviewer's suggestions, we ran additional experiments to compare the total training time of Coded-PrivateML with [26] (on the same Amazon EC2 setting of [26], by measuring both offline and online time, on a synthesized dataset as in [26, Table 2] with 10000 data points and 1000 features). We observed that CodedPrivateML with $N = 50$ clients achieves $20.9\times$ speedup in the total training time against the OT-based approach of [26], while the scalability of [26] is limited to 2 parties. We will include these results with additional experiments in our revised version.

2) [*Memory consumption and communication costs*] We summarize the communication and storage cost per client in Table 1 where $m, d, J$ denote the number of data points, features, and iterations, respectively. We will include this table in our revised version.

Table 1: Complexity Analysis.

|       | Comm.        | Storage     |
|-------|--------------|-------------|
| Data  | $O(mdN/K)$   | $O(md/K)$   |
| Model | $O(dNJ)$     | $O(dN)$     |

[Meta-Review · NeurIPS 2020]

The initial reviews showed some disagreement about this paper, with two positive reviewers noting the reduction in computational and communication costs compared to prior solutions, and two more negative reviewers with some concerns in particular regarding novelty and comparison with respect to previous work. After reading the author rebuttal and further discussion, the doubts regarding the comparison to recent work were lifted, leading to one reviewer increasing his/her score. While some concerns remain regarding the applicability of the work to non-linear models, the merits of the work are judged significant enough, and we decided the paper should be accepted. In the final version, the authors are asked to be more explicit about the potential limitations of the degree-1 approximation to the sigmod, and to add a discussion about how one may go about extending the approach to more complicated (deep) models.